# Topological electronic structure and spin texture of quasi-one-dimensional higher-order topological insulator Bi$_4$Br$_4$

Wenxuan Zhao[1], Ming Yang[2,3], Runzhe Xu[1], Xian Du[1], Yidian Li[1], Kaiyi Zhai[1], Cheng Peng[4], Ding Pei[4], Han Gao[5], Yiwei Li[5], Lixuan Xu[1], Junfeng Han[6,7], Yuan Huang[6,7], Zhongkai Liu[5,8], Yugui Yao[6,7], Jincheng Zhuang[2,3], Yi Du[2,3]✉, Jinjian Zhou[6]✉, Yulin Chen[4,5,8]✉ & Lexian Yang[1,9,10]✉

The notion of topological insulators (TIs), characterized by an insulating bulk and conducting topological surface states, can be extended to higher-order topological insulators (HOTIs) hosting gapless modes localized at the boundaries of two or more dimensions lower than the insulating bulk. In this work, by performing high-resolution angle-resolved photoemission spectroscopy (ARPES) measurements with submicron spatial and spin resolution, we systematically investigate the electronic structure and spin texture of quasi-one-dimensional (1D) HOTI candidate Bi$_4$Br$_4$. In contrast to the bulk-state-dominant spectra on the (001) surface, we observe gapped surface states on the (100) surface, whose dispersion and spin-polarization agree well with our ab-initio calculations. Moreover, we reveal in-gap states connecting the surface valence and conduction bands, which is a signature of the hinge states inside the (100) surface gap. Our findings provide compelling evidence for the HOTI phase of Bi$_4$Br$_4$. The identification of the higher-order topological phase promises applications based on 1D spin-momentum locked current in electronic and spintronic devices.

The bulk-boundary correspondence connecting the boundary modes to the bulk topology is a central paradigm of topological quantum materials[1–3]. A prime example is the three-dimensional (3D) topological insulator (TI), where two-dimensional (2D) gapless boundary modes protected by time reversal symmetry exist on the surfaces of 3D insulating bulk (Fig. 1a, left). This fundamental property is described by the Z$_2$ topological invariants. Prominently, the concept of TI can be extended to 'higher order' with more elaborative classification by Z$_4$ invariants[4–6]. By contrast to the bulk-surface correspondence in 3D TI, higher-order topological insulators (HOTIs) host gapless modes (e.g. 1D topological hinge states) at the boundaries of two or more dimensions lower than the 3D bulk (Fig. 1a, right)[7–11]. Based on the spin-momentum locked hinge states with low dimensionality and protection by nontrivial topology, HOTIs have provided significant scientific implication and application potential due to their novel properties, such as dissipationless transport, efficient charge-to-spin conversion,

[1]State Key Laboratory of Low Dimensional Quantum Physics, Department of Physics, Tsinghua University, Beijing 100084, China. [2]School of Physics, Beihang University, Beijing 100191, China. [3]Centre of Quantum and Matter Sciences, International Research Institute for Multidisciplinary Science, Beihang University, Beijing 100191, China. [4]Department of Physics, Clarendon Laboratory, University of Oxford, Parks Road, Oxford OX1 3PU, UK. [5]School of Physical Science and Technology, ShanghaiTech University and CAS-Shanghai Science Research Center, Shanghai 201210, China. [6]Centre for Quantum Physics, Key Laboratory of Advanced Optoelectronic Quantum Architecture and Measurement (MOE), School of Physics, Beijing Institute of Technology, Beijing 100081, China. [7]Yangtze Delta Region Academy of Beijing Institute of Technology, Jiaxing 314001 Zhejiang province, China. [8]ShanghaiTech Laboratory for Topological Physics, Shanghai 200031, China. [9]Frontier Science Center for Quantum Information, Beijing 100084, China. [10]Collaborative Innovation Center of Quantum Matter, Beijing, China. ✉e-mail: yi_du@buaa.edu.cn; jjzhou@bit.edu.cn; yulin.chen@physics.ox.ac.uk; lxyang@tsinghua.edu.cn

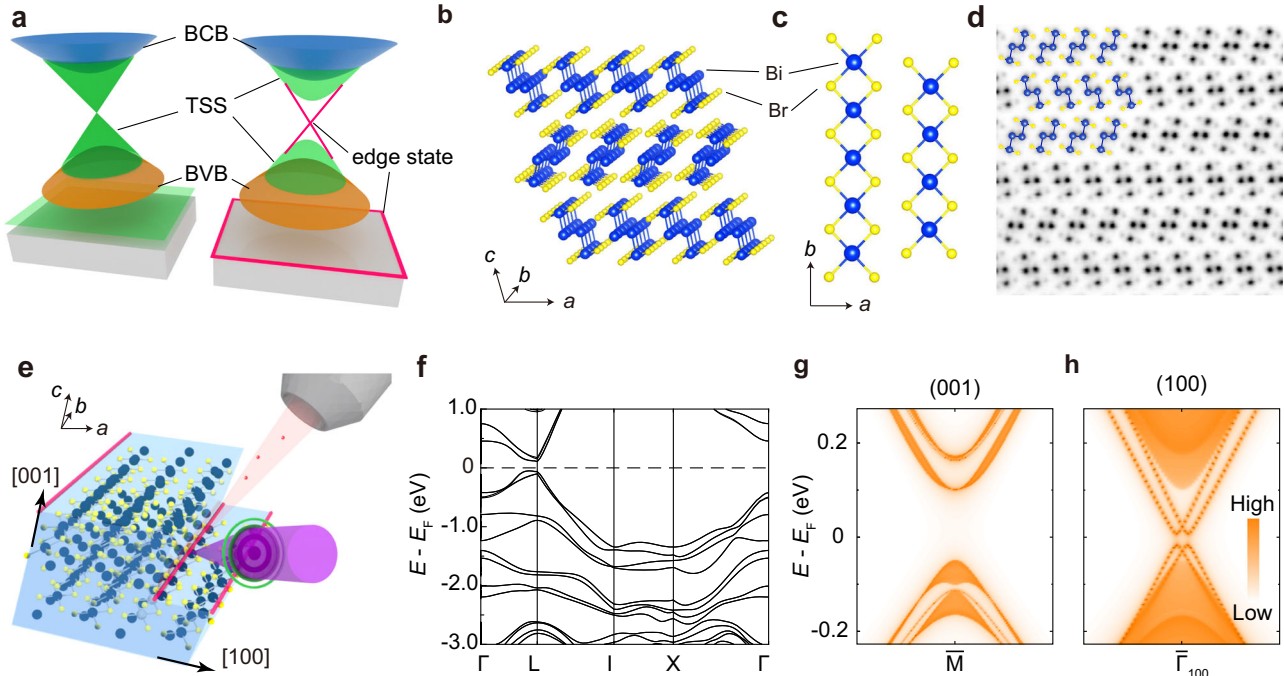

**Fig. 1 | Higher-order topological electronic properties and crystal structure of Bi$_4$Br$_4$. a** Schematic of electronic structures of Z$_2$ topological insulators (TIs, left) and higher-order TIs (HOTIs, right). BCB: bulk conduction band; TSS: topological surface states; BVB: bulk valence band. **b, c** The crystal structure of Bi$_4$Br$_4$ from the approximately side view and the top view, respectively. **d** Cross-section image with atomic resolution measured with scanning transmission electron microscope (STEM). The crystal structure viewed along *b* is overlaid. **e** Schematic illustration of μ-ARPES measurements on the (001) and (100) surfaces of Bi$_4$Br$_4$ single crystal with hinge states along the edges of terraces. **f** Ab-initio calculation of the electronic structures of Bi$_4$Br$_4$. **g, h** (001) and (100) surface-projected band structures calculated along the chain direction, respectively.

possession of Majorana zero modes for topological quantum computation, and possible realization of spin-triplet superconductivity[12–15].

Up to date, many materials have been predicted to be HOTIs, such as Bi$_{2-x}$Sm$_x$Se$_3$, $X$Te$_2$ ($X$ = Mo, W), EuIn$_2$As$_2$, MnBi$_{2n}$Te$_{3n+1}$, and several 2D materials, et al.[11,16–29]. However, experimental investigations on these materials[30] remain inadequate and elusive. For example, while the existence of hinge states in the single crystalline bismuth has been established by scanning tunneling spectroscopy (STM) and Josephson-interference measurements, the theoretically predicted topological electronic structure distinguishing the HOTI phase has not been observed experimentally[13]. The metallic property of bismuth also makes it challenging to investigate and apply the hinge states. Therefore, it is still highly demanding to experimentally search for more feasible HOTIs.

Recently, a new type of quasi-1D crystal Bi$_4$Br$_4$ has aroused extensive research interest in this field[31–38]. It crystallizes in a base-centered monoclinic structure with lattice constants $a = 13.065$ Å, $b = 4.338$ Å, $c = 20.062$ Å, and $\beta = 107.4°$ (space group $C2/m$)[39,40] (Fig. 1b, c). Bi$_4$-Br$_4$ chains extending along the *b* axis form Bi$_4$Br$_4$ layers in the *ab* plane. The monolayer Bi$_4$Br$_4$ has been identified as a 2D quantum spin Hall (QSH) insulator with a large band gap of about 180 meV[34–36], and bulk Bi$_4$Br$_4$ can be viewed as AB stacked monolayers along the *c* axis with the adjacent monolayers rotating 180° with respect to each other, as shown in Fig. 1b. Figure 1d presents the cross-section image measured using scanning transmission electron microscope with atomic-resolution, confirming the crystal structure and suggesting the high quality of our samples.

## Results and discussion
### Basic band structure of Bi$_4$Br$_4$
Bulk Bi$_4$Br$_4$ features a two-fold rotational anomaly that induces a topological crystalline insulator phase involving a higher-order bulk-boundary correspondence[38,41]. A pair of helical 1D gapless modes,

known as hinge states, exist at the crosslines of the (001) and (100) surfaces, realizing a HOTI phase[42,43]. After cleavage, the sample surface naturally exposes a large number of terraces and hinges. Both the surface states and hinge states can be probed by angle-resolved photoemission spectroscopy (ARPES) with sub-micron spatial resolution (μ-ARPES or nano-ARPES)[44–46], as schematically shown in Fig. 1e. Our ab-initio calculations reveal a nonzero $Z_4$ topological invariant and a bulk band gap of about 180 meV near the *L* point, as shown in Fig. 1f[34,35]. According to our surface-projected calculations, there are gapped surface states emerge as two branches of linear-like bands that split along $k_y$ with an energy gap of about 25 meV on the (100) surface, in drastic contrast to the absence of surface states on the (001) surface (Fig. 1g, h).

Experimentally, non-trivial transport measurements provided clues for the facet-dependent surface states[32,47], and scanning tunneling spectroscopy measurements revealed QSH edge states on monolayer steps[31,33]. However, despite the previous ARPES reports of trails of hinge states[32] that were limited by the spatial and/or energy resolutions, direct observation of the gapped (100) surface states and the hinge states of Bi$_4$Br$_4$ as well as their spin textures, is still lacking, which makes the HOTI phase of Bi$_4$Br$_4$ tentative and inconclusive. In this work, by performing high-resolution ARPES measurements with sub-micron spatial and spin resolutions, we systematically investigate the electronic structure of Bi$_4$Br$_4$. On the (001) surface, we observe only the bulk band without evidence for surface states. On the (100) surface, remarkably, we observe gapped surface states with a large splitting related to two unpinned Dirac points[42] that were not unraveled experimentally in the previous studies. We also reveal the spin-momentum locking character of the gapped (100) surface states, in good agreement with ab-initio calculations. Moreover, there exist electronic states inside the (100) surface gap, suggesting the existence of hinge states. Our results provide compelling evidence for the HOTI phase, which makes Bi$_4$Br$_4$ an ideal material platform for exploring the electronic properties and application potential of 1D boundary modes.

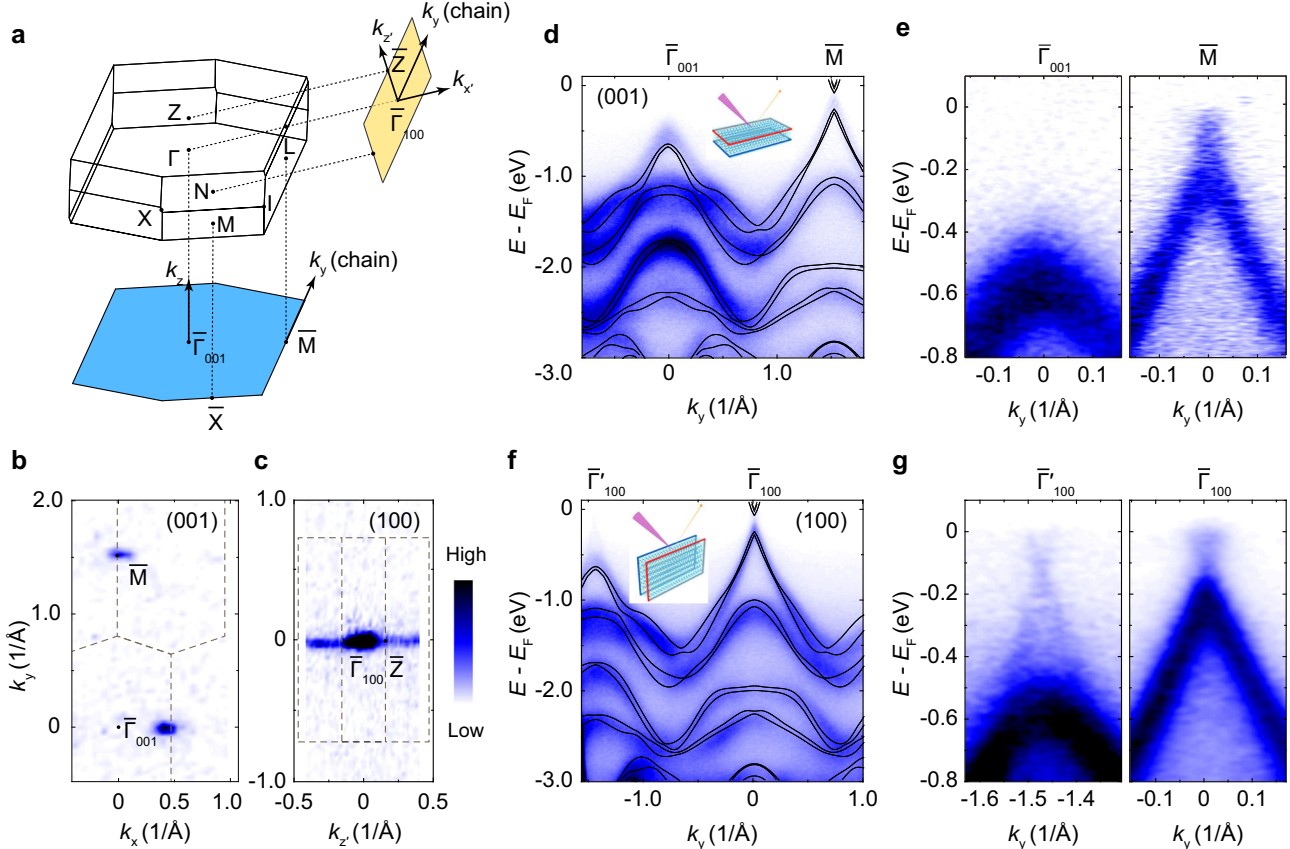

**Fig. 2 | Overall band structures of Bi$_4$Br$_4$ measured on the (001) and (100) surfaces. a** Bulk and surface Brillouin zone with high-symmetry points indicated. **b, c** Fermi surface measured on the (001) and (100) surfaces. **d** Band dispersions along the chain direction on the (001) surface with ab-initio calculated results overlaid. The inset shows the schematics of ARPES measurements on the (001) surface. **e** Zoom-in plot of the fine band structures around the $\bar{\Gamma}_{001}$ (left) and $\bar{M}$ (right) points. **f** Band dispersions along the chain direction measured on the (100) surface with ab-initio calculated results overlaid. The inset shows the schematics of ARPES measurements on the (100) surface. **g** Zoom-in plot of the fine band structures around the $\bar{\Gamma}_{100}$ (right) and $\bar{\Gamma}'_{100}$ (left) points on the (100) surface. Data were collected with synchrotron-based nano-ARPES at $hv = 101$ eV at 20 K.

By performing synchrotron-based nano-ARPES, intrinsic electronic structures measured on both the (001) and (100) surfaces are shown in Fig. 2. Figure 2a shows the bulk Brillion zone (BZ) of Bi$_4$Br$_4$ and its surface projections. On the (001) surface, the Fermi surface consists of point-like features at the $\bar{M}$ point (Fig. 2b). By contrast, there exists an extra line-like feature along $\bar{\Gamma}_{100}\bar{Z}$ on the (100) surface (Fig. 2c). In Fig. 2d we show the band dispersion along the chain direction in a broad energy and momentum range. We observe a hole band with the band top near $E_F$ - 0.6 eV and $E_F$ - 0.25 eV around the $\bar{\Gamma}_{001}$ and $\bar{M}$ points respectively (Fig. 2e). The experimental band structures are in good agreement with the ab-initio calculations of the bulk states (Fig. 2d). For direct comparison, the band dispersions detected on the (100) surface are shown in Fig. 2f, g. While the bulk bands are similar to those measured on the (001) surface, there exist linear dispersions near $E_F$, which lead to the quasi-1D line in the constant-energy contours in Fig. 2c. The linear bands are in good accordance with our (100) surface-projected ab-initio calculations in Fig. 1h. Based on the observations above, we identify the linear bands that only emerge on the side surface as the (100) surface states.

**Characteristic electronic structure of the HOTI phase**

The HOTI phase of Bi$_4$Br$_4$ is characterized by the gapped (100) surface states and hinge states inside the surface gap. Figure 3a shows our ab-initio calculations on a 10-layer Bi$_4$Br$_4$ ribbon with a width of 30 chains in the $a$-axis direction. Prominently, inside the surface band gap, there exist 1D topological hinge states protected by the two-fold rotational symmetry $2_{[010]}$[42]. Both the gapped (100) surface states and the hinge

states originate from the QSH edge states of the monolayer Bi$_4$Br$_4$. In view of the real space, when Bi$_4$Br$_4$ monolayers stack along the $c$ axis to form the bulk crystal, the QSH edge states of adjacent monolayers become non-degenerate due to the AB stacking sequence. Their quantum hybridization[31,34,37] open an energy gap of tens of meV, resulting in the splitting gapped (100) surface states. Moreover, the QSH edge states at the two sides of each monolayer hybridize with those from different neighboring monolayers[31,37] (Fig. 3b). As a result, only two gapless modes can survive at the hinges of bulk Bi$_4$Br$_4$ (red lines in Fig. 3b), which are precisely the hinge states.

To scrutinize the gapped (100) surface states and hinge states, we utilize laser-based μ-ARPES with superb energy and momentum resolutions to overcome the limitation of synchrotron-based nano-ARPES in Fig. 2 and investigate the fine electronic structure of Bi$_4$Br$_4$ near $E_F$[44]. Prominently, we observe pivotal spectral features of the gapped surface states on the (100) surface (Fig. 3c–e), in drastic contrast to the bulk states measured on the (001) surface (Supplementary Fig. 3) but in excellent agreement with our calculation in Fig. 1h. On the one hand, both the calculated and measured valence/conduction bands of the (100) surface states present a splitting-like feature along the $k_y$ direction. On the other hand, both the measured and calculated (100) surface states exhibit a band gap although the experimental value (peak-to-peak gap of 40 meV and leading-edge gap of 28 meV) is slightly larger than the calculated one (25 meV). The band splitting and the surface gap can be observed more evidently in the stacked energy distribution curves (EDCs) of the band dispersion across the $\bar{\Gamma}_{100}$ point (Fig. 3f). We emphasize that neither the splitting nor the band gap of

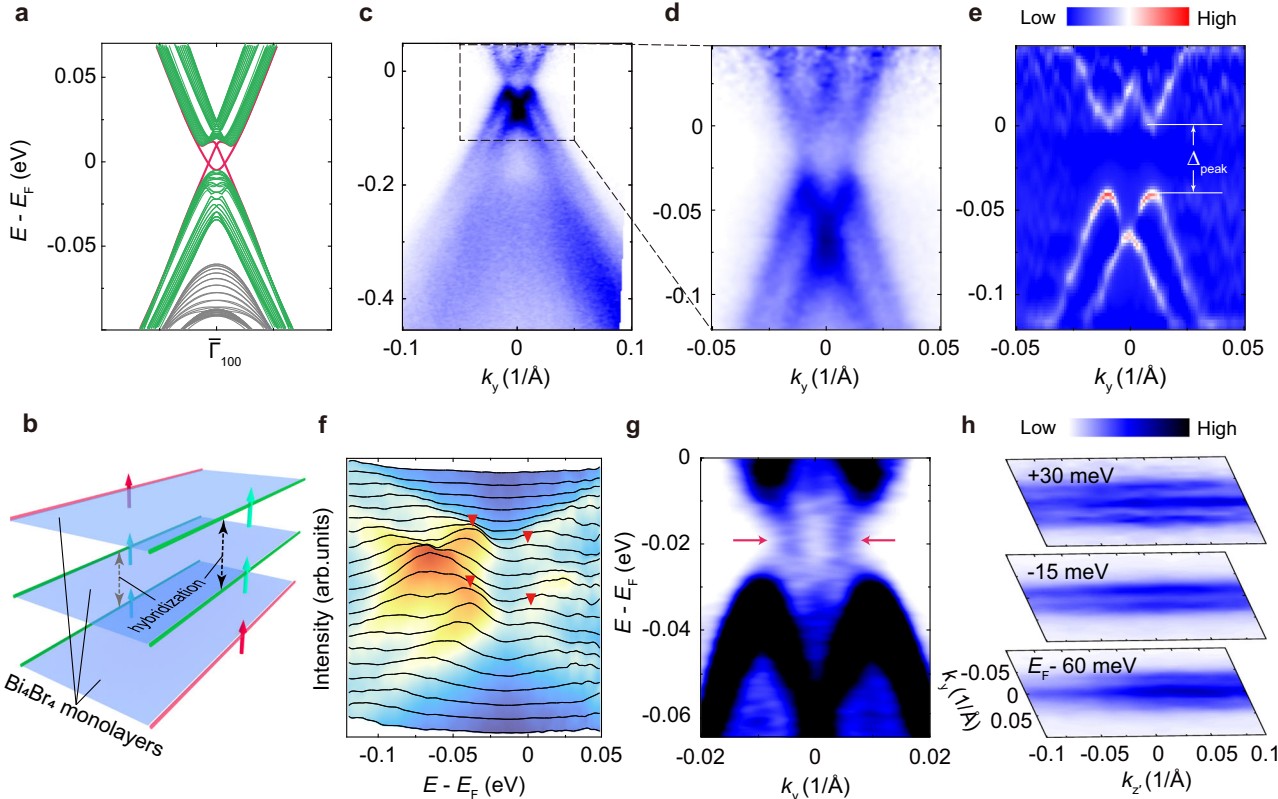

**Fig. 3 | Gapped (100) surface states and in-gap states. a** Ab-initio calculation of the electronic structures of a 10-layer $Bi_4Br_4$ slab. **b** Schematic illustration of the quantum hybridization between quantum spin Hall edge states from adjacent $Bi_4Br_4$ monolayers. **c, d** Fine band structure measured using laser-based μ-ARPES on the (100) surface. Data were collected at 80 K and divided by the Fermi-Dirac function for better comparison between the experimental and calculated conduction band. **e** Curvature of the spectra in **d** showing the (100) surface band gap.

The peak-to-peak gap is about 40 meV as indicated. **f** Stacking plot of energy distribution curves (EDCs) of the spectra in **d**. The red triangles mark the peak positions of the conduction and valence bands. **g** ARPES spectra after deconvolution to remove the spectral broadening effects. Data were collected at 30 K. The spectrum without deconvolution is shown in the Supplementary Information. **h** Constant energy contours at selected binding energies.

the (100) surface states has been discovered in previous ARPES studies[32].

More importantly, there exist extra electronic states inside the gap of the (100) surface bands. These in-gap states can be better resolved in the ARPES spectra after deconvolution, which is a commonly used method to remove the spectral broadening effect (Fig. 3g). Notably, the dispersion of the in-gap states are clearly resolved inside the (100) surface gap, in line with the calculated hinge states in Fig. 3a. The main features above can also be observed in the constant energy contours (CECs) at different binding energies as well (Fig. 3h). Inside the gap, straight line-like spectral features are revealed near $E_F$-15 meV, indicating the 1D characteristic of the in-gap states along the chain direction. Above or below the surface gap region, the CECs exhibit separated line-like contours, corresponding to the (100) surface states. We emphasize that our spectral measurements on the (100) surface show nice agreement with the theory of the HOTI phase of $Bi_4Br_4$. More spectroscopic evidences for the existence of in-gap states are presented in the Supplementary Information, including the direct visualization of the edge states using scanning tunneling spectroscopy (see supplementary Figs. 2, 3, and 4).

**Spin texture of the surface states**

The spin texture of the boundary states is a more compelling characteristic of topological quantum materials. As shown in Fig. 4a, b, our calculations of the (100) surface states show a high spin polarization along the $z'$ direction (see the Brillouin zone in Fig. 2a), while the spin polarization along the surface normal ($x'$) is much weaker and the spin

polarization along the y direction is nearly zero at $k_{z'} = 0$ (see supplementary Fig. 5). The calculated spin texture of the (100) surface states is shown in Fig. 4c, d at selected binding energies. To further confirm the HOTI phase of $Bi_4Br_4$, we detect the spin textures of the gapped surface states and hinge states by performing laser-based spin-resolved μ-ARPES[44] on the (100) surface of $Bi_4Br_4$. Fig. 4e, f show the measured $z'$- and $x'$-component of the spin-polarization of the band dispersion along the chain direction. The left and right branches of valence and conduction surface bands show opposite polarizations for both $z'$ and the surface normal ($x'$) components, where the spin polarization is much weaker along the surface normal direction, in good agreement with the calculated results. Fig. 4g, h shows the spin-polarized MDCs and the $z'$-component of the spin polarization. The experimental $z'$-component of the spin polarization ratio is about 40% for both the conduction and valence bands of the surface states. The experimental spin-momentum locking property of the (100) surface states is in good agreement with the ab-initio calculations[48] (Fig. 4a, b).

It is worth noting that the electronic states inside the surface gap (around $E_F$ - 15 meV) show no spin-polarization in our data. While the hinge states show a spin-momentum locking property in the calculation (Supplementary Fig. 6), it is extremely challenging to resolve the spin-polarized hinge states using spin-ARPES due to their weak intensity and limited resolutions of spin-ARPES. On the other hand, spin-ARPES accumulates signals from hinge states propagating along different edges (depending on the thickness of the terraces[37]), which may cancel out the spin-polarization of the hinge states in spin-ARPES measurements. Nevertheless, considering the homology (QSH edge

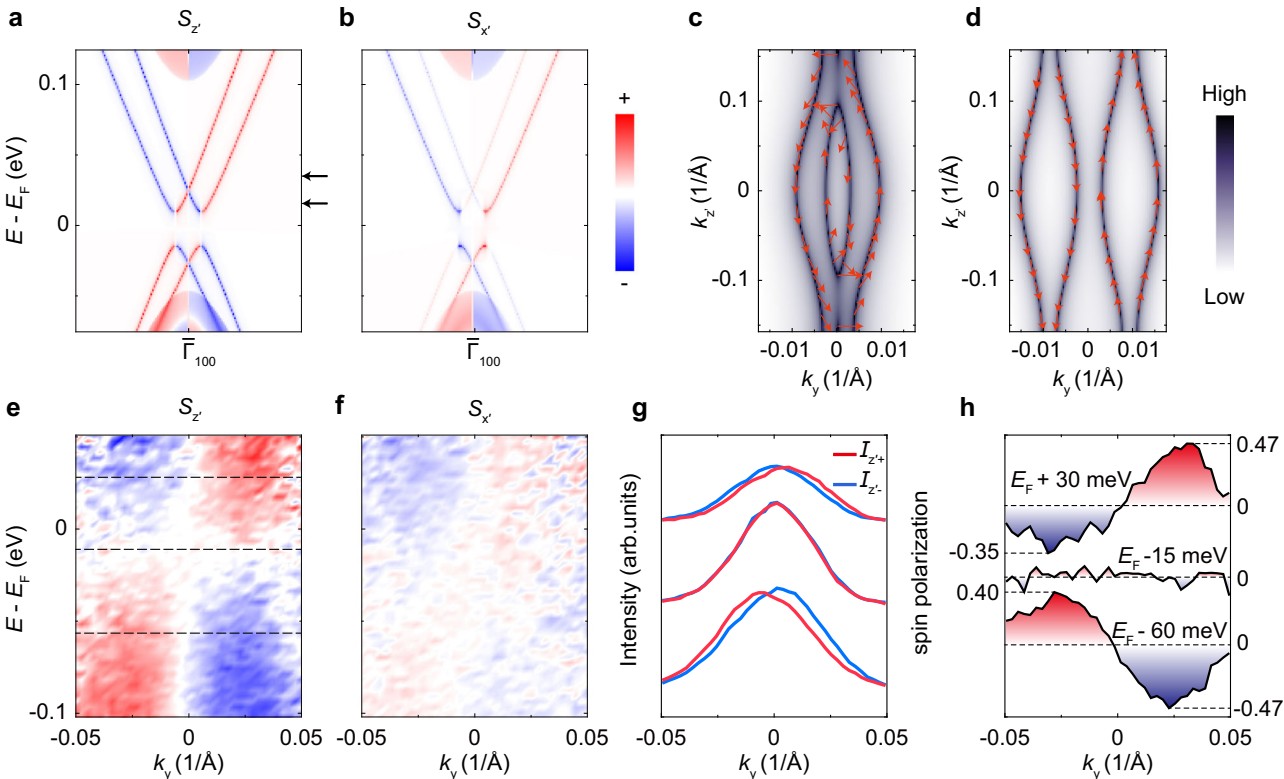

**Fig. 4 | Spin textures of the (100) surface states of Bi$_4$Br$_4$. a, b** Calculated $z'$- and $x$'-component of spin polarization of the (100) surface states. **c, d** Calculated spin textures of the (100) surface states at the energy positions marked by the arrows in **a**. **e, f** Experimental $z'$- and $x'$-component of spin polarization of the (100) surface states. **g, h** $z'$-component of spin-resolved momentum distribution curves (MDCs) and corresponding spin polarization at selected binding energies marked by the dashed lines in **e**. Data were collected using a 7-eV laser at 140 K.

states of monolayer Bi$_4$Br$_4$ before stacking into bulk) of the (100) surface states and hinge states, our consistent observation of the spin texture of the side surface states also suggests the spin-momentum locking property of the in-gap hinge states.

To conclude, we present the characteristic electronic structure and spin texture of the gapped (100) surface states of Bi$_4$Br$_4$ together with the signature of the existence of hinge states, which provides compelling evidence for the HOTI phase of Bi$_4$Br$_4$. Hosting 1D gapless modes at the hinges, the system generalizes the category of topological quantum materials possessing highly directional spin currents[42,49], where the 2D QSH insulators/3D weak TIs with their topological edge/surface states have widely stimulated the research interests[50–52]. We expect that our identification of the HOTI phase of Bi$_4$Br$_4$ will motivate further investigations on higher-order topological phases and the gapless hinge modes to pursue the promising prospect of electronic and spintronic devices.

## Methods

### Sample growth and characterization
High-quality Bi$_4$Br$_4$ single crystals were synthesized using solid-state reaction method. High-purity reagents Bi and BiBr$_3$ powders with equal molar ratio were thoroughly mixed under Ar atmosphere and then sealed in a quartz tube under a vacuum below $1 \times 10^{-5}$ mbar. The quartz tube was placed in a two-zone furnace with the temperature gradient from 558 K to 461 K for 72 h. After natural cooling down, single crystals with a size larger than $2 \times 0.2 \times 0.1$ mm$^3$ were nucleated at the high-temperature side of the quartz tube. X-ray diffraction measurements (Panalytical Aeris) were performed with a Cu K$\alpha$ radiation source to determine the single structure of Bi$_4$Br$_4$ at room temperature. X-ray photoemission spectra were measured using a Scienta Omicron photoelectron spectrometer equipped with a monochromatic Al K$_{\alpha1}$

radiation ($h\nu = 1486.7$ eV) under a high vacuum below $2 \times 10^{-9}$ mbar. The samples were further characterized using the high-angle annular dark-field scanning transmission electron microscope (HAADF-STEM) measurements. A TEM lamella was first prepared using a Zeiss Cross-beam 550 FIB-SEM and then measured on a probe and image-corrected FEI Titan Themis Z microscope that is equipped with a hot-field emission gun working at 300 kV. The temperature-dependent resistivity was measured using a physical properties measurement system (PPMS, Quantum Design).

### Ab-initio calculations
First-principles calculations were carried out to investigate the electronic properties and topological band character of bulk Bi$_4$Br$_4$ with experimental lattice parameters, using the Vienna ab-initio simulation package[53]. We utilized the Heyd–Scuseria–Ernzerhof hybrid functional (HSE06) to describe the exchange-correlation potential[54], and set the energy cutoff of the plane-wave basis to 300 eV. To construct maximally localized Wannier functions (MLWFs) for the $p$-orbitals of Bi and Br atoms, we used the WANNIER90 code[55] and performed the calculations on a $6 \times 6 \times 3$ $k$-mesh. From these MLWFs, we built ab-initio tight-binding models to compute the electronic structures of Bi$_4$Br$_4$ ribbons. Additionally, we calculated the surface electronic structures and their spin polarizations by combining the ab-initio tight-binding models with surface Green functions method[34,56].

### Synchrotron-based nano-ARPES measurements
The nano-ARPES measurements were conducted under ultra-high vacuum (UHV) better than $1 \times 10^{-10}$ mbar at the BL07U endstation of Shanghai Synchrotron Radiation Facility (SSRF). Bi$_4$Br$_4$ single crystals were cleaved in situ to expose (100) and (001) surfaces. The synchrotron beam was focused using a Fresnel zone plate and a spatial

resolution better than 400 nm was achieved. The measurements were performed at 20 K with a photon energy of 101 eV. The data were collected with a Scienta DA30L electron analyzer. The total energy and angular resolutions were set to 35 meV and 0.2°, respectively.

### Laser-based μ-ARPES measurements with spin resolution
Laser-based μ- and spin-resolved ARPES measurements were conducted at Tsinghua University[44]. The 7-eV laser was generated by frequency doubling in a KBBF crystal and focused by an optics lens to reach a sub-micron spatial resolution. The samples were cleaved in situ under UHV better than $5 \times 10^{-11}$ mbar. The data of the (001) and (100) surfaces were directly measured on the correspondingly cleaved samples without the observation of mixed domains. The typical size of the samples is around $0.2 \times 2 \times 0.5$ mm and the typical size of the terraces measured by micro-ARPES 2D scan is around one hundred microns along the chain direction and several microns perpendicular to the chain direction. ARPES data were collected by Scienta DA30L electron analyzer. The total energy and angular resolutions were set to 1.8 meV and 0.2°, respectively. Therefore, the energy and momentum resolution of the laser-ARPES data is much better than the synchrotron-based data. The spin-resolved data was taken by a spin detector (Ferrum detector, Focus GmbH) based on very low energy electron diffraction (VLEED) on a surface-passivated Fe/W(100) target that is attached behind the Scienta DA30L analyzer. The energy resolution of the spin-resolved measurements was about 22 meV.

## Data availability
The data sets that support the findings of this study are available from the corresponding author upon request.

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

## Acknowledgements

This work is funded by the National Key R&D Program of China (Grant No. 2022YFA1403201, 2022YFA1403100, and 2022YFA1403400) and the National Natural Science Foundation of China (No. 12274251, 12104039, 62275016, 12074021, and 12274016). L.Y. acknowledges support from the Tsinghua University Initiative Scientific Research Program and the Fund of Science and Technology on Surface Physics and Chemistry Laboratory (No. XKFZ202102). Y.C. acknowledges the support from the Oxford-ShanghaiTech collaboration project and the Shanghai Municipal Science and Technology Major Project (grant 2018SHZDZX02). Y.D. thanks for the support through the Fundamental Research Funds for the Central Universities (YWF-22-K–101).

## Author contributions

L.Y. conceived the experiments. W.Z. carried out ARPES measurements with the assistance of R.X., X.D., Y.L., K.Z., C.P., D.P., H.G., Y.L., L.X. and Z.L. Ab-initio calculations were performed by J.Z. and X.D. contributed to the preliminary DFT calculations. Single crystals were synthesized and characterized by J.H., Y.H., Y.Y., M.Y., J.Z. and Y.D. The paper was written by W.Z., L.Y. and Y.C. All authors contributed to the scientific planning and discussion.

## Competing interests

The authors declare no competing interests.
