## [Peer Review File · Nature Communications]

REVIEWER COMMENTS

Reviewer #1 (Remarks to the Author):

In "Topological electronic structure and spin texture of quasi-one-dimensional higher-order topological insulator Bi₄Br₄" the authors perform a combined theoretical DFT and ARPES study of the higher-order topological insulator candidate Bi₄Br₄.

The main experimental finding is the observation of gapped surface states at the (100) surface, inside which in-gap states are appearing.

The authors claim that the presence of such hinge states provides compelling evidence for the predicted higher-order topological insulator phase in Bi₄Br₄.

While I do certainly appreciate the observation of in-gap 1D modes by angle resolved photoemission spectroscopy, I do not agree with the authors that per se this proves the topological crystalline insulating phase of the material. There are three different reasons.

1) First, as acknowledged by the authors, the electronic states inside the surface gap do not show any spin polarization. As the authors certainly will be aware, the hinge states of time-reversal symmetric higher-order topological insulators must be helical and therefore possess one-dimensional spin-momentum locking.

2) Second, even when considering that the absence of spin polarization can be due to the fact that the data accumulate signals from different hinges (and thus assuming the hinge states to be actually helical) the experiments do not demonstrate that the hinge states are anomalous. I would like to recall that the hinge modes of a HOTI have the same characteristics of the edge states of a quantum spin Hall insulator: an odd number of time-reversal related pairs of hinge states are present. On the contrary, an even number of time-reversal related pairs of hinge modes signals a topological trivial phase. I am not sure that this even-odd difference can be detected with the experiments performed by the authors.

3) Third, the higher-order topological insulator phase of Bi₄Br₄ is protected by a twofold rotation symmetry. Therefore, at the rotational invariant surface the rotational anomaly first discussed by Liang Fu and Chen Fang must be present [see Science Advances 5, 2019]. This implies that two surface Dirac cones at unpinned point are expected on the 010 surface. The authors do not show spectroscopic and/or transport signatures of the presence of such Dirac cones, which I believe is important to provide evidence of the higher-order topological insulator phase of Bi₄Br₄.

Reviewer #2 (Remarks to the Author):

The manuscript by W. X. Zhao et al, presents an ARPES study of Bi₄Br₄, a material that is supposed to be a higher order topological insulator (HOTI) with non trivial hinged states at the edges between (100) and (001) terraces.

The same material was studied about two years ago by the group of Takeshi Kondo (Noguchi et al., Nature Materials 20 473 (2021)) who provided first experimental evidence that the electronic structure would be in line with the theoretical prediction.

Compared to the early work of Noguchi et al, the ARPES data of Zhao et al. are clearer. On the (100) surface the authors resolve pairs of gapped Dirac cones in agreement with DFT calculations. Inside the gap the authors claim to have identified traces of the hinged states. If this was correct, the present work would provide “compelling evidence for the HOTI phase of Bi₄Br₄” as stated by the authors in the abstract and would be a significant step beyond the previous efforts.

However, I do not see evidence for hinged states in their data. The residual spectral weight inside the gap seems fully consistent with a simple superposition of the tails of conduction and valence surface states.

In the manuscript the authors do not explain how the dispersion of the hinged states was determined (yellow bars in Fig.3(g)). By looking at the individual MDCs it looks like the marked positions were arbitrary chosen to match the theoretical expectations.

I did not understand where the improvement of data quality comes from. Similarly to Noguchi et al. they used synchrotron based nano-ARPES and a micro focus laser system to acquire their data. It is not at all clear what the decisive factor is, whether it is spatial resolution, energy resolution, sample quality... In the manuscript they vaguely refer to the limitations of synchrotron-based nano-ARPES, but those are not obvious even for an expert reader. This point should be clearly addressed in the main text.

The manuscript lacks essential information on how the experiment was performed. The authors measured the electronic structure on two orthogonal faces of Bi₄Br₄, which is technically not easy, and do not explain how this was achieved. This is an important piece of information because it is very rare for a single crystal to have two stable orthogonal cleavage planes. Were data from the (100) and the (001) surfaces measured on the same sample? How large were the probed terraces? Any information in this direction would improve the manuscript.

Reviewer #3 (Remarks to the Author):

The manuscript by Zhao et al. presents a detailed photoemission study on an interesting crystalline material, namely Bi_4Br_4 , predicted to be a so-called higher-order topological insulator (HOTI). Probably the most prominent electronic feature of a quasi-one dimensional (1D) HOTI is the presence of hinge states that appear inside a band gap of the (2D) surface states. The data in this paper have been obtained from high resolution laser photoemission and synchrotron-radiation based μ -ARPES. The data are of excellent quality and certainly represent a significant methodological advance compared to previous experiments in the literature. Furthermore, the paper contains spin-resolved data and respective spin-resolved band-structure calculations which are essential for the understanding and interpretation of the experimental results. The manuscript is well written, concise and contributes to the research on a highly topical class of materials.

However, I have some concerns regarding the reliability of the analysis that is done to experimentally prove the existence of the signature feature, namely the hinge states. There is only Fig. 3g where the reader can immediately see two dispersing (apparently roughly parabolic) bands formed by yellow bars, indicating the "peak positions of the MDCs" (momentum distribution curves). In the curvature plot Fig. 3e nothing of that is discernible. One understands that these measurements are extremely demanding, but since the strong claim of the paper depends on this analysis, a more convincing presentation is necessary. Particular in view of the finite energy resolution (compared to a gap with of 40 meV) or due to some surface inhomogeneities some overlapping tails or background intensities of the approaching sets of bands above and below might cause the in-gap intensity artificially. Furthermore, one might wonder about the lack of noise in the shown MDC (3g) and the method to extract the peak positions therein. It would be highly welcome if the authors could rule out and discuss possible artifacts in this analysis.

Response to Reviewer #1:

Reviewer's Comment:

In "Topological electronic structure and spin texture of quasi-one-dimensional higher-order topological insulator Bi₄Br₄" the authors perform a combined theoretical DFT and ARPES study of the higher-order topological insulator candidate Bi₄Br₄. The main experimental finding is the observation of gapped surface states at the (100) surface, inside which in-gap states are appearing. The authors claim that the presence of such hinge states provides compelling evidence for the predicted higher-order topological insulator phase in Bi₄Br₄. While I do certainly appreciate the observation of in-gap 1D modes by angle resolved photoemission spectroscopy, I do not agree with the authors that per se this proves the topological crystalline insulating phase of the material. There are three different reasons.

Authors' response:

We thank reviewer #1 for appreciating our experimental observations. While we understand the concern of the reviewer and appreciate his/her suggestions on more solid confirmations, we would like to point out that the higher-order topological insulator (HOTI) phase of Bi₄X₄ (X = I, Br) is also characterized by the gapped (100) surface states and the in-gap hinge states [Sci. Adv. 4, eaat0346 (2018); 2D Mater. 6, 031004 (2019); arXiv:2005.14710]. ARPES measurements of gapped side surface states and gapless hinge states, along with their agreement with *ab-initio* calculations have been widely used to verify the HOTI phase of Bi₄X₄ but with much limited data quality in previous reports [Nat. Mater. 20, 473–479 (2021); Phys. Rev. X 11, 031042 (2021)]. With more experiments and data analysis, we present our point-by-point response to the reviewer's comments below:

Reviewer's Comment:

1) First, as acknowledged by the authors, the electronic states inside the surface gap do not show any spin polarization. As the authors certainly will be aware, the hinge states of time-reversal symmetric higher-order topological insulators must be helical and therefore possess one-dimensional spin-momentum locking.

Authors' response:

We thank the reviewer for this comment and fully agree that the hinge states should possess one-dimensional spin-momentum locking. **Indeed, our *ab-initio* calculation (supplementary Fig. 6), for the first time, theoretically presented the nontrivial spin texture of the hinge states.** Considering the homology of the hinge states and the (100) surface states (both originate from QSH edge states of monolayer

Bi₄Br₄), the observed spin texture of the (100) surface states and its agreement with our *ab-initio* calculation support the spin-momentum locking property of the hinge states. Nevertheless, it is extremely difficult to directly measure the spin texture of the hinge states (actually, to our best knowledge, there is no report of spin polarization of any topological states at 1D edge measured by spin-ARPES up to date). The lack of spin polarization in our experiment, as explained in our manuscript, does not deny the spin-momentum locking of the hinge states. It is mainly due to three reasons: (a) The spin-momentum locking property of the hinge state depends on the thickness of the step [arXiv:2005.14710]. After cleavage, there are many different hinges with opposite spin-momentum locking properties. The summation of the photoelectrons from these hinges induces in-gap signals without spin-polarization. A similar result has been reported in Nat. Mater. 20, 473–479 (2021). (b) The photoemission intensity of the hinge states is much weaker than that of the surface states. It is already difficult to measure the hinge states using normal ARPES. Considering that the detection efficiency of the spin-ARPES is about one order lower than the normal ARPES, it is extremely difficult to measure the hinge states using spin-ARPES. (c) The momentum and energy resolutions of our spin-ARPES experiment are more limited than the laser-based micro-ARPES. The signal of the hinge states, though possibly polarized, may smear out by the broadening effects.

Reviewer's Comment:

2) Second, even when considering that the absence of spin polarization can be due to the fact that the data accumulate signals from different hinges (and thus assuming the hinge states to be actually helical) the experiments do not demonstrate that the hinge states are anomalous. I would like to recall that the hinge modes of a HOTI have the same characteristics of the edge states of a quantum spin Hall insulator: an odd number of time-reversal related pairs of hinge states are present. On the contrary, an even number of time-reversal related pairs of hinge modes signals a topological trivial phase. I am not sure that this even-odd difference can be detected with the experiments performed by the authors.

Authors' response:

We thank the reviewer for this comment. We agree with the reviewer that the number of edge states is important for demonstrating the HOTI phase of Bi₄Br₄. We would like to point out that the number of edge states in quantum materials has never been revealed by spectroscopic measurements. For example, in ZrTe₅, scanning tunneling microscopy (STM) experiments confirmed the existence of edge states but nothing was known about the number of edge states on the experimental level [Nat Commun 12, 406 (2021)]; The edge

states of the quantum anomalous Hall states in Cr-doped $(\text{Bi, Sb})_2\text{Te}_3$ was visualized by microwave impedance microscopy (MIM) without knowing their number [PNAS, 116, 14511 (2019)]; Previous STM measurements also revealed the existence of edge states in Bi_4Br_4 . But the number of edge states was not experimentally resolved [Nat. Mater. 21, 1111–1115 (2022)].

On the other hand, we emphasize that there is only one pair of edge modes in the monolayer Bi_4Br_4 , as confirmed by the *ab-initio* calculation [Nano Lett. 14, 8, 4767–4771 (2014)]. When the bulk Bi_4Br_4 crystal is formed by stacking the Bi_4Br_4 layers, the adjacent edge states hybridize to form the gapped (100) surface states, leaving gapless modes along certain hinges. According to the *ab-initio* calculations, there is, if any, only one pair of hinge states along each hinge. The configuration of the hinge modes depends on the terminations of the sample surface, as shown in Fig. R1.

Fig.R1 | Energy dispersion and distribution of the hinge states in Bi_4Br_4 . [adopted from arXiv:2005.14710]

Reviewer's Comment:

3) Third, the higher-order topological insulator phase of Bi_4Br_4 is protected by a twofold rotation symmetry. Therefore, at the rotational invariant surface the rotational anomaly first discussed by Liang Fu and Chen Fang must be present [see Science Advances 5, 2019]. This implies that two surface Dirac cones at unpinned point are expected on the 010 surface. The authors do now show spectroscopic and/or transport signatures of the presence of such Dirac cones, which I believe is important to provide evidence of the higher-order topological insulator phase of Bi_4Br_4 .

Authors' response:

We thank the reviewer for this discussion. It has been predicted that Bi_4Br_4 features a two-fold rotational anomaly that induces a topological crystalline insulator (TCI) phase involving a higher-order bulk-boundary correspondence [Science Advances 5, eaat2374 (2019); 2D Mater. 6 031004 (2019)] and hosting 1D gapless modes at hinges. It would be straightforward to measure the “unpinned” Dirac cones at the (010) surface to experimentally identify the TCI phase, and furthermore, the rotational anomaly of Bi_4Br_4 . Nevertheless, it is extremely challenging to cleave the crystals along the (010) surface, making it impossible to directly measure the “unpinned” Dirac cones by ARPES. Nevertheless, the HOTI phase of Bi_4Br_4 , as the main topic of our work, is manifested by the gapped surface states on the (100) surface and the in-gap hinge states, which have been adopted by many experimental and theoretical investigations to identify the HOTI phase [Nat. Mater. 20, 473–479 (2021); Phys. Rev. X 11, 031042 (2021); Sci. Adv. 4, eaat0346 (2018); 2D Mater. 6, 031004 (2019); arXiv:2005.14710]. Our conclusion of evidence for the HOTI phase of Bi_4Br_4 and the advancement compared to previous works in this field is based on the following experimental and theoretical results:

(1) Our *ab-initio* calculation, in good agreement with previous results [2D Mater. 6 031004 (2019); Nat. Phys. 15, 470–476 (2019)], confirms the HOTI phase of Bi_4Br_4 with characteristic electronic and spin structures of both the gapped (100) surface states and the in-gap hinge states. The perfect agreement between our experiment and the calculation on both the (001) and (100) surfaces, together with the observation of the in-gap hinge states, is a strong evidence for the HOTI phase of Bi_4Br_4 .

(2) For the first time, our experiment presents the spin texture of the (100) surface states, and our *ab-initio* calculation reveals the non-trivial spin texture of the hinge states. Considering the homology of the (100) surface states and hinge states (both originate from the QSH edge states of monolayer Bi_4Br_4 before stacking into bulk), our consistent observation of the spin texture of the side surface states suggests the anomaly of the hinge states. This part of our work completes the experimental identification of the HOTI phase of Bi_4Br_4 .

(3) Compared with previous ARPES experiments that claimed the HOTI phase of Bi_4Br_4 system [Nat. Mater. 20, 473–479 (2021)], our data has unprecedented data quality. The characteristic surface gap on the (100) surface remains undetected in previous works and the identification of the gapless boundary modes is challenged due to the mixture with signals from the side surface [Nat. Mater. 21, 1111–1115 (2022)]. By distinguishing the gapped (100) surface states and in-gap hinge states respectively, we believe our work

resolves the most characteristic electronic structure and provides the strongest spectroscopic evidence for the HOTI phase of Bi_4Br_4 up to date.

In summary, we appreciate the helpful discussion with reviewer #1. While his/her proposal of studying the electronic structure on the (010) surface is interesting and important, it is technically impossible at present. In practice, even the side cleavage to expose the (100) surface is already difficult. We hope to obtain a clean and flat (010) surface in the future to conduct the experiment proposed by reviewer #1. We would like to emphasize that our work is a combined investigation by ARPES and *ab-initio* calculation (the new insights provided by our *ab-initio* calculation deserve more attention), which provides a comprehensive investigation and new perspective on the electronic structure of Bi_4Br_4 . The electronic structure and spin texture of the (100) surface states and the in-gap hinge states are the most characteristic properties of the HOTI phase that can be experimentally accessed so far. Our experiment is of the highest quality and the only one that shows the fine details with perfect agreement with *ab-initio* calculations up to date, which we believe is important for the community. To relieve the concern of reviewer #1, we have added more discussion about the anomaly of the Bi_4Br_4 hinge states and cited the corresponding literature. We believe our results merit the requirement of Nature Communications and hope the reviewer can reconsider his/her recommendation.

Response to Reviewer #2:

Reviewer's Comment:

The manuscript by W. X. Zhao et al, presents an ARPES study of Bi₄Br₄, a material that is supposed to be a higher order topological insulator (HOTI) with non trivial hinged states at the edges between (100) and (001) terraces.

The same material was studied about two years ago by the group of Takeshi Kondo (Noguchi et al., Nature Materials 20 473 (2021)) who provided first experimental evidence that the electronic structure would be in line with the theoretical prediction.

Compared to the early work of Noguchi et al, the ARPES data of Zhao et al. are clearer. On the (100) surface the authors resolve pairs of gapped Dirac cones in agreement with DFT calculations. Inside the gap the authors claim to have identified traces of the hinged states. If this was correct, the present work would provide “compelling evidence for the HOTI phase of Bi₄Br₄” as stated by the authors in the abstract and would be a significant step beyond the previous efforts.

Authors' response:

We thank reviewer #2 for acknowledging that our ARPES data are “clearer” and potentially would be “a significant step beyond the previous efforts”. Below are our point-by-point responses to the reviewer's comments/questions:

Reviewer's Comment:

However, I do not see evidence for hinged states in their data. The residual spectral weight inside the gap seems fully consistent with a simple superposition of the tails of conduction and valence surface states.

Authors' response:

We thank the reviewer for carefully reading our manuscript and pointing out the possible artifact in the origin of the in-gap signal. We argue that the spectral weight inside the gap cannot be fully explained by the superposition of the tails of conduction and valence surface states (although the tails indeed partially contribute to the in-gap signal). Below we provide more evidence of the existence of the hinge states based on the following experimental facts:

- 1. We observe spectral features that cannot be explained by the tails of the surface states but consistent with extra in-gap states.** To visualize the in-gap states better, we deconvolute ARPES spectra to remove the thermal broadening and experimental resolution effects, which is a commonly used method

[see, e.g., Phys. Rev. Lett. 110, 047004 (2013)]. As shown in Fig. R2a, we observe extra states inside the (100) surface gap. In particular, the in-gap states have non-trivial dispersions in good agreement with the calculated hinge states (Fig. R2b and d). The in-gap states can be further visualized by normalizing the deconvoluted spectra with the integrated energy distribution curve (EDC) (Fig. R2c). The superposition of the surface state tails cannot explain these dispersing in-gap states. Moreover, our detailed check of the momentum distribution curves (MDCs) in the gap region also provides spectral features that are consistent with the intrinsic in-gap states (please see Fig. R6 and our response to the next point raised by the reviewer).

Fig.R2 | Spectroscopic evidence for the existence of in-gap states. **a**, ARPES spectra after deconvolution to remove the spectral broadening effects. **b**, The same as **a** but with guide to eyes for the dispersion of the in-gap states. **c**, Deconvoluted ARPES spectra in **a** normalized by the integrated energy distribution curve (EDC). **d**, *Ab-initio* calculation of the electronic structures of a 10-layer Bi_4Br_4 slab (the same as Fig. 3a in the main text). The red and green curves represent the hinge states and (100) surface states, respectively.

2. The in-gap states show different polarization dependence from that of the (100) surface states. In principle, photons with linear-horizontal (LH) or linear-vertical (LV) polarization give symmetric ARPES spectral weight along high-symmetry momentum direction. However, if the photon polarization is off LH or LV, the matrix element effect can induce asymmetric spectral weight [J. Electron. Spectros. Relat. Phenomena, 214, 29-52, (2017)]. This spectral asymmetry may be different for the surface and hinge states due to the complex configuration of sample boundaries (such as hinges). As presented in Fig. R3, we observe asymmetric spectral weight of the ARPES spectra with the incident photon polarization in 45° with respect to the horizon. The intensity ratio between the left and right branches of the surface states is about 1.6 for both the surface valence and conduction bands as presented by the peaks of the momentum distribution curves. If the in-gap signal were simply caused by the broadening of the surface states, the two peaks of the in-gap MDC will show a similar intensity ratio. However, the in-gap MDC clearly exhibits a

less asymmetric intensity ratio (about 1.3) than both the valence and conduction surface states, which means the in-gap intensity is not simply the tails of the gapped surface states. Thus, a natural explanation of the in-gap states would be the intrinsic signal of the hinge states.

Fig. R3 | Asymmetry of ARPES spectra measured using 45°-polarized photons. **a**, ARPES spectra measured using 7 eV laser with 45° linear polarization at 80 K. **b-d**, MDCs at selected binding energies showing different ratios between the intensities of the left and right branches of the spectra. The energy positions are marked in **a**. The red and green curves are the fitting results with two Lorentz peaks (gray curves) and a constant background. The dashed lines mark the intensity of the peaks for direct comparison.

3. The in-gap states can be observed also on the (001) surface. The hinges are shared by the (001) and (100) surfaces while there are no surface states on the (001) surface. As discussed in Supplementary Fig.2 (replotted in Fig. R4), we observe additional states on the (001) surface, which appear inside the gap of the (100) surface states, as manifested by a shoulder in the EDC (see the arrow in Fig. R4c). These states cannot be induced by the mixed signal from the (100) domains since the more intensive conduction and valence bands of the (100) surface states were not observed. Moreover, our micro-ARPES with sub-micron spatial resolution excludes the mixture of the (100) domains on the (001) cleaved surface (no gapped surface states have been observed on the (001) cleaved samples). Therefore, the additional states on the (001) surface is a piece of evidence for the existence of the hinge states. **Please note that a similar argument has been used in [Nat. Mater. 20, 473–479 (2021)] to identify the hinge states detected on the (001) surface.** Our comparison of the gapless modes detected on the (001) surface and the gapped (100) surface states excludes the ambiguity of the mixed (100) signal, which further solidifies the confirmation of the HOTI phase of Bi₄Br₄.

Fig.R4 | Comparison between laser-based μ -ARPES measurements on the (001) and (100) surfaces.

a, b, ARPES spectra along the chain direction measured on the (100) and (001) surfaces of Bi_4Br_4 , respectively. **c,** Comparison between the EDCs at $k_y = 0$ (integrated with a momentum range of 0.03 1/\AA) measured on the (100) (blue line) and (001) (orange line) surfaces. The blue dashed line is the (100) EDC divided by the Fermi-Dirac function.

4. Our scanning tunneling spectroscopy (STS) measurements directly visualize the edge states. The

detection of edge states can be directly performed by scanning tunneling microscopy, supporting the non-trivial topological phase of Bi_4Br_4 [Nat. Mater. 21, 1111–1115 (2022)]. We perform STS measurements on the (001) surface of Bi_4Br_4 (unfortunately, the STS experiment on the (100) surface is not possible at present). Fig. R5a shows the topography near the edges on the (001) surface. In the STS map, we clearly observe enhanced density of states at both edges of the terrace near the Fermi level, which directly visualizes in-gap edge states.

Fig.R5 | STS measurements on the edge state of Bi_4Br_4 . **a,** Topography of the (001) surface edge of Bi_4Br_4 . **b,** The dI/dV map detected at the Fermi energy with the same range of **a**.

Reviewer's Comment:

In the manuscript the authors do not explain how the dispersion of the hinged states was determined (yellow bars in Fig.3(g)). By looking at the individual MDCs it looks like the marked positions were arbitrary chosen to match the theoretical expectations.

Authors' response:

Fig. R6 | MDC signature of the in-gap states. Left: stacking plot of MDCs near the gap region. The blue triangles, yellow bars, and gray circles mark the peak positions of the MDCs. The dashed lines are a guide of eyes for the band dispersion of the hinge states. Right: zoom-in plot of the MDCs inside the (100) surface gap.

We thank the reviewer for carefully reviewing our manuscript. The markers in the MDCs are not arbitrary. They indicate the features of the MDCs. As shown in the zoom-in plot of the MDCs in the gap regions in Fig. R6, each yellow bar corresponds to either a peak or a shoulder in the MDCs. While the determination of the positions of the yellow bar is by naked eyes, they are above the noise level of the data and must indicate meaningful characters of the intrinsic signal. The multi-peak structure of the MDCs in the gap region cannot be explained by the tails of the surface states but is consistent with the existence of in-gap hinge states. We would also like to emphasize that the MDC analysis is consistent with the deconvoluted ARPES spectra in Fig. R2 (please also see our response to the first point raised by the reviewer).

Reviewer's Comment:

I did not understand where the improvement of data quality comes from. Similarly to Noguchi et al. they used synchrotron based nano-ARPES and a micro focus laser system to acquire their data. It is not at all clear what the decisive factor is, whether it is spatial resolution, energy resolution, sample quality... In the

manuscript they vaguely refer to the limitations of synchrotron-based nano-ARPES, but those are not obvious even for an expert reader. This point should be clearly addressed in the main text.

Authors' response:

We thank the reviewer for the suggestion on adding detailed information about the experiments. The factors regarding the improvement of our data quality are included in our revised article, i.e. the following three aspects:

1. In the main text of Noguchi et al.'s work [Nat. Mater. 20, 473–479 (2021)], they provide ARPES data detected only on the (001) cleavage surface. Since there are no surface states on the (001) surface of Bi_4Br_4 , the signals mostly consist of the bulk band and residual intensity possibly from hinges. The gapped (100) surface states can hardly be detected in this configuration. By contrast, our synchrotron- and laser-based ARPES measurements of the gapped surface states were directly conducted on the (100) cleavage surface (Fig.2 and Fig.3 in the main text, also see our response to the next point raised by the reviewer) thus a fine electronic structure of the gapped (100) surface states and hinge states can be observed.

2. Noguchi et al. also tried to provide ARPES data on the (100) surface of Bi_4Br_4 using synchrotron-based nano-ARPES in their Supplementary Information but with limited data quality. In comparison, the improvement of our synchrotron-based data is attributed to the better energy and spatial resolutions in the nano-ARPES station in SSRF with respect to that of the photoelectron microscopy station in Elettra (where Noguchi et al. performed nano-ARPES measurements on the (100) surface).

3. For ARPES measurements of Bi_4Br_4 , laser-ARPES provides much better data quality due to the enhanced energy and momentum resolution compared to the synchrotron-based ARPES. In particular, we built a powerful laser-based AREPS system with sub-micron spatial resolution and high-efficiency spin detection [Rev. Sci. Instrum. 94, 023903 (2023)], which is perfectly suitable for the measurements of Bi_4Br_4 . **We would like to remind that Noguchi et al. did not conduct laser-based ARPES measurement on the (100) surface of Bi_4Br_4 .**

4. We spend plenty of time improving the quality of our single crystals and the cleavage of our samples. To obtain high-quality samples with large side surface, we have synthesized many batches of samples and carefully characterized their structures. While it is not difficult to cleave along the (100) surface, a large, clean, and flat (100) surface is not guaranteed in each cleavage. We have measured more than 50 samples to get the highest data quality and repeat the results.

Per the suggestions of the reviewer, we have added the sentence “Therefore, the energy and momentum resolution of the laser-ARPES data is much better than the synchrotron-based data.” in line 22, page 13 to provide more experimental details in the revised manuscript.

Reviewer’s Comment:

The manuscript lacks essential information on how the experiment was performed. The authors measured the electronic structure on two orthogonal faces of Bi₄Br₄, which is technically not easy, and do not explain how this was achieved. This is an important piece of information because it is very rare for a single crystal to have two stable orthogonal cleavage planes. Were data from the (100) and the (001) surfaces measured on the same sample? How large were the probed terraces? Any information in this direction would improve the manuscript.

Authors’ response:

We thank the reviewer for this discussion. The reviewer is right that “it is very rare for a single crystal to have two stable orthogonal cleavage planes”. However, for the quasi-one-dimensional crystals with weak inter-chain coupling such as ZrTe₅, Bi₄I₄, and Bi₄Br₄, it is technically possible to cleave the crystals along two different directions [Nat Commun. 12:406 (2021); Phys. Rev. X 11, 031042 (2021)].

Our data from the (100) and the (001) surfaces were measured on different samples but well repeated. It is technically impossible to cleave the same sample along different directions since the samples were glued on the sample holder along either the (001) or (100) direction. Previous work reported mixed (001) and (100) domains on the (001) cleavage surface of Bi₄I₄. However, in our laser-ARPES experiments with sub-micron spatial resolution, we did NOT observe the (100) surface states on the (001) cleavage surface of Bi₄Br₄. In fact, it is naturally expected to get a better sample surface for ARPES measurements by cleaving the desired surface.

The typical size of the samples is around $0.2 \times 2 \times 0.5$ mm. The typical size of terraces measured by micro-ARPES 2D scan is around one hundred microns along the chain direction and several microns perpendicular to the chain direction in both experiments on the (001) and (100) surfaces.

Per the suggestion of the reviewer, we have added more information about sample cleavage in the revised manuscript, i.e. “The samples were cleaved *in-situ* under UHV better than 5×10^{-11} mbar. The data of the (001) and (100) surfaces were directly measured on the correspondingly cleaved samples without the observation of mixed domains. The typical size of the samples is around $0.2 \times 2 \times 0.5$ mm and the typical

size of the terraces measured by micro-ARPES 2D scan is around one hundred microns along the chain direction and several microns perpendicular to the chain direction.” in line 16, page 13.

We hope our point-by-point responses to the reviewer’s comments/questions can relieve his/her concerns and look forward to his/her recommendation for the publication of our manuscript.

Response to Reviewer #3:

Reviewer's Comment:

The manuscript by Zhao et al. presents a detailed photoemission study on an interesting crystalline material, namely Bi₄Br₄, predicted to be a so-called higher-order topological insulator (HOTI). Probably the most prominent electronic feature of a quasi-one dimensional (1D) HOTI is the presence of hinge states that appear inside a band gap of the (2D) surface states. The data in this paper have been obtained from high resolution laser photoemission and synchrotron-radiation based μ -ARPES. The data are of excellent quality and certainly represent a significant methodological advance compared to previous experiments in the literature. Furthermore, the paper contains spin-resolved data and respective spin-resolved band-structure calculations which are essential for the understanding and interpretation of the experimental results. The manuscript is well written, concise and contributes to the research on a highly topical class of materials.

Authors' response:

We thank reviewer #3 for acknowledging that our data are “of excellent quality and certainly represent a significant methodological advance compared to previous experiments in the literature”, our spin-resolved data are “essential for the understanding and interpretation of the experimental results”, and “the manuscript is well written, concise and contributes to the research on a highly topical class of materials”. We appreciate his/her helpful discussions that greatly help improve our manuscript. Below are our point-by-point responses to the reviewer's comments/questions:

Reviewer's Comment:

However, I have some concerns regarding the reliability of the analysis that is done to experimentally prove the existence of the signature feature, namely the hinge states. There is only Fig. 3g where the reader can immediately see two dispersing (apparently roughly parabolic) bands formed by yellow bars, indicating the "peak positions of the MDCs" (momentum distribution curves). In the curvature plot Fig. 3e nothing of that is discernible. One understands that these measurements are extremely demanding, but since the strong claim of the paper depends on this analysis, a more convincing presentation is necessary. Particular in view of the finite energy resolution (compared to a gap with of 40 meV) or due to some surface inhomogeneities some overlapping tails or background intensities of the approaching sets of bands above and below might cause the in-gap intensity artificially.

Authors' response:

We thank the reviewer for this comment. The signal of hinge states is contributed by finite number of one-dimensional edges, whose intensity is much weaker than the signal of the (100) surface states. They can be resolved only in the MDCs (see below). The curvature plot was obtained by calculating the curvature along the energy axis, which is not sensitive to the weak intensity in the small energy range (the gap is about 40 meV). Below we provide more analysis to rule out possible artifacts and give more evidence for the existence of in-gap hinge states (please also see our response to reviewer #2's first comment):

1. We observe spectral features that cannot be explained by the tails of the surface states but consistent with extra in-gap states. To visualize the in-gap states better, we deconvolute ARPES spectra to remove the thermal broadening and experimental resolution effects, which is a commonly used method [see, e.g., Phys. Rev. Lett. 110, 047004 (2013)]. As shown in Fig. R7a, we observe extra states inside the (100) surface gap. In particular, the in-gap states have non-trivial dispersions in good agreement with the calculated hinge states (Fig. R7b and d). The in-gap states can be further visualized by normalizing the deconvoluted spectra with the integrated energy distribution curve (EDC) (Fig. R7c). The superposition of the surface state tails cannot explain these dispersing in-gap states. Moreover, our detailed check of the momentum distribution curves (MDCs) in the gap region also provides spectral features that are consistent with the intrinsic in-gap states (please see Fig. R11 and our response to the next point raised by the reviewer).

Fig.R7 (the same as Fig. R2) | Spectroscopic evidence for the existence of in-gap states. **a**, ARPES spectra after deconvolution to remove the spectral broadening effects. **b**, The same as **a** but with guide to eyes for the dispersion of the in-gap states. **c**, Deconvoluted ARPES spectra in **a** normalized by the integrated energy distribution curve (EDC). **d**, *Ab-initio* calculation of the electronic structures of a 10-layer Bi_4Br_4 slab (the same as Fig. 3a in the main text). The red and green curves represent the hinge states and (100) surface states, respectively.

2. The in-gap states show different polarization dependence from that of the (100) surface states. In principle, photons with linear-horizontal (LH) or linear-vertical (LV) polarization give symmetric ARPES spectral weight along high-symmetry momentum direction. However, if the photon polarization is off LH or LV, the matrix element effect can induce asymmetric spectral weight [J. Electron. Spectros. Relat. Phenomena, 214, 29-52, (2017)]. This spectral asymmetry may be different for the surface and hinge states due to the complex configuration of sample boundaries (such as hinges). As presented in Fig. R8, we observe asymmetric spectral weight of the ARPES spectra with the incident photon polarization in 45° with respect to the horizon. The intensity ratio between the left and right branches of the surface states is about 1.6 for both the surface valence and conduction bands as presented by the peaks of the momentum distribution curves. If the in-gap signal were simply caused by the broadening of the surface states, the two peaks of the in-gap MDC will show a similar intensity ratio. However, the in-gap MDC clearly exhibits less asymmetric intensity ratio (about 1.3) than both the valence and conduction surface states, which means the in-gap intensity is not simply the tails of the gapped surface states. Thus, a natural explanation of the in-gap states would be the intrinsic signal of the hinge states.

Fig. R8 (the same as Fig. R3) | Asymmetry of ARPES spectra measured using 45° -polarized photons. **a**, ARPES spectra measured using 7 eV laser with 45° linear polarization at 80 K. **b-d**, MDCs at selected binding energies showing different ratios between the intensities of the left and right branches of the spectra. The energy positions are marked in **a**. The red and green curves are the fitting results with two Lorentz peaks (gray curves) and a constant background. The dashed lines mark the intensity of the peaks for direct comparison.

3. The in-gap states can be observed also on the (001) surface. The hinges are shared by the (001) and (100) surfaces while there are no surface states on the (001) surface. As discussed in Supplementary Fig.2 (replotted in Fig. R9), we observe additional states on the (001) surface, which appear inside the gap of the

(100) surface states, as manifested by a shoulder in the EDC (see the arrow in Fig. R9c). These states cannot be induced by the mixed signal from the (100) domains since the more intensive conduction and valence bands of the (100) surface states were not observed. Moreover, our micro-ARPES with sub-micron spatial resolution excludes the mixture of the (100) domains on the (001) cleaved surface (no gapped surface states have been observed on the (001) cleaved samples). Therefore, the additional states on the (001) surface is a piece of evidence for the existence of the hinge states. **Please note that a similar argument has been used in [Nat. Mater. 20, 473–479 (2021)] to identify the hinge states detected on the (001) surface.** Our comparison of the gapless modes detected on the (001) surface and the gapped (100) surface states excludes the ambiguity of the mixed (100) signal, which further solidifies the confirmation of the HOTI phase of Bi_4Br_4 .

Fig.R9 (the same as Fig. R4) | Comparison between laser-based μ -ARPES measurements on the (001) and (100) surfaces. **a, b**, ARPES spectra along the chain direction measured on the (100) and (001) surfaces of Bi_4Br_4 , respectively. **c**, Comparison between the EDCs at $k_y = 0$ (integrated with a momentum range of 0.03 1/\AA) measured on the (100) (blue line) and (001) (orange line) surfaces. The blue dashed line is the (100) EDC divided by the Fermi-Dirac function.

4. Our scanning tunneling spectroscopy (STS) measurements directly visualize the edge states. The detection of edge states can be directly performed by scanning tunneling microscopy, supporting the non-trivial topological phase of Bi_4Br_4 [Nat. Mater. 21, 1111–1115 (2022)]. We perform STS measurements on the (001) surface of Bi_4Br_4 (unfortunately, the STS experiment on the (100) surface is not possible at present). Fig. R10a shows the topography near the edges on the (001) surface. In the STS map, we clearly observe enhanced density of states at both edges of the terrace near the Fermi level, which directly visualizes in-gap edge states.

Fig.R10 (the same as Fig. R5) | STS measurements on the edge state of Bi_4Br_4 . **a**, Topography of the (001) surface edge of Bi_4Br_4 . **b**, The dI/dV map detected at the Fermi energy with the same range of **a**.

We have added more evidence in our revised article to rule out the possible artifacts in contrast to the hinge states signals based on our new data and analysis.

Reviewer’s Comment:

Furthermore, one might wonder about the lack of noise in the shown MDC (3g) and the method to extract the peak positions therein. It would be highly welcome if the authors could rule out and discuss possible artifacts in this analysis.

Authors’ response:

The MDCs and EDCs in Fig. 3 of the main text seem smooth since (a) the raw data were collected by integrating the signal for a long time to obtain high-quality data; (b) the MDCs and EDCs were integrated in certain energy or momentum regions. (c) The noise of the data hides behind the stacking plot of the curves. In fact, as shown in Fig. R11 (the same as Fig. R6), there is still noise in the MDCs in the region where intrinsic signals are weak. But the features marked by the yellow bars are above the noise level of the data.

Fig. R11 (the same as Fig. R6) | MDC signature of the in-gap states. Left: stacking plot of MDCs near the gap region. The blue triangles, yellow bars, and gray circles mark the peak positions of the MDCs. The dashed lines are a guide of eyes for the band dispersion of the hinge states. Right: zoom-in plot of the MDCs in the gap region.

We hope our point-by-point responses to the reviewer's questions can relieve his/her concerns and look forward to his/her recommendation for the publication of our manuscript.

List of changes

1. We added “Bulk Bi_4Br_4 features a two-fold rotational anomaly that induces a topological crystalline insulator phase involving a higher-order bulk-boundary correspondence. A pair of helical 1D gapless modes, known as hinge states, exist at the crosslines of the (001) and (100) surfaces, realizing a HOTI phase.” in line 7, page 13 to emphasize the previous theoretical works on the TCI and HOTI phase of Bi_4Br_4 .
2. We added “Notably, the MDCs exhibit extra features as manifested by the peaks or shoulders (indicated by yellow bars in Fig. 3g), which suggests the existence of in-gap states.” in line 21, page 5 to explain the determination of the positions of the yellow bars.
3. We added “More importantly, the in-gap states show a dispersion that agrees well with the calculation, and can be resolved in ARPES spectra after deconvolution (Supplementary Note 2).” in line 23, page 5 to emphasize the agreement deconvoluted ARPES spectra and calculation on the hinge states.
4. We added “More spectroscopic evidences for the existence of in-gap states are presented in the Supplementary Information, including the direct visualization of the edge states using scanning tunneling spectroscopy (see supplementary Fig. 2, 3, and 4).” in line 5, page 6 to remind additional evidence on the in-gap states shown in the Supplementary Information.
5. We added “Nevertheless, considering the homology (QSH edge states of monolayer Bi_4Br_4 before stacking into bulk) of the (100) surface states and hinge states, our consistent observation of the spin texture of the side surface states also suggests the spin-momentum locking property of the in-gap hinge states.” in line 5, page 7 to argue for the spin polarization of the hinge states.
6. We added information on the details of the sample cleavage surfaces, namely “The samples were cleaved *in situ* under UHV better than 5×10^{-11} mbar. The data of the (001) and (100) surfaces were directly measured on the correspondingly cleaved samples without the observation of mixed domains.” And “The typical size of the samples is around $0.2 \times 2 \times 0.5$ mm and the typical size of the terraces measured by micro-ARPES 2D scan is around one hundred microns along the chain direction and several microns perpendicular to the chain direction.” in line 16, page 13.
7. We added information regarding the improvement of our data quality in line 22, page 13 “Therefore, the energy and momentum resolution of laser-ARPES data is much better than the synchrotron-based data.”
8. We added the Supplementary Note “Spectroscopic evidence for the existence of in-gap states” in the Supplementary Information.

9. We revised the Supplementary Note “Comparison between laser-based μ -ARPES measurements on the (001) and (100) surfaces.” in the Supplementary Information to clarify its vindication for the HOTI phase of Bi_4Br_4 .

10. We added the Supplementary Note “Visualization of the edge states STS measurements on the (100) surface of Bi_4Br_4 using scanning tunneling spectroscopy.” in Supplementary Information.

11. We updated the reference list.

Reviewers' comments:

Reviewer #1 (Remarks to the Author):

The authors have adequately replied to my concerns.

They acknowledge that the spin texture as well as the number of hinge modes cannot be firmly proven experimentally. On the other hand, they point out that in combination with their density functional theory calculations, compatibility between the observed hinge modes the anomalous ones of a higher-order topological insulator can be established.

I agree on this point.

As it concerns the study at the (010) surface the authors simply pointed out that at present it is technically impossible to perform such study.

I have no other remarks on the study.

Reviewer #2 (Remarks to the Author):

The authors revised the manuscript following the referees comments. They added some additional information on how the experiment was performed and included in Supplementary Information a deconvolution of the ARPES data to support the identification of in-gap states on the (100) surface.

The result is only marginally different from what was shown in Fig.3(d) of the main text. Without the guide-to-the-eye and the theoretical prediction of in-gap states it would be very difficult to identify these additional features from the ARPES data alone.

The authors explain in the reply that the positions of the yellow bars marking the signature of in-gap states were defined by naked eyes, considering peaks and shoulders that are above the noise level. Clearly this is not a unique method to extract the dispersion: (i) there were other peaks and shoulders in the MDCs that the authors ignored; (ii) given that the MDCs were integrated in a certain energy window (reply to referee #3), one may wonder to what extent those shallow peaks were affected by the integration.

I don't think that the authors could have done much better with the existing data set, but I have the impression that the "compelling evidence" for hinged states isn't actually that strong.

Reviewer #3 (Remarks to the Author):

The authors have carefully considered the comments and questions from the reviewer. Most of the unclear statements or conclusions in the first version of the manuscript have been clarified by additional figures and discussions. The results are certainly at the experimental limit at the moment and ask for further experimental investigations in the future. However, the paper represents a significant step towards a better understanding of this interesting material. I recommend to publish it in Nat. Commun. in its present form.

Response to Reviewer #1:

Reviewer's Comment:

The authors have adequately replied to my concerns.

They acknowledge that the spin texture as well as the number of hinge modes cannot be firmly proven experimentally. On the other hand, they point out that in combination with their density functional theory calculations, compatibility between the observed hinge modes the anomalous ones of an higher-order topological insulator can be established.

I agree on this point.

As it concerns the study at the (010) surface the authors simply pointed out that at present it is technically impossible to perform such study.

I have no other remarks on the study.

Authors' response:

We thank reviewer #1 for helping us improve the manuscript. We are grateful to his/her agreement on our identification of the HOTI phase.

Response to Reviewer #2:

We thank reviewer #2 for his/her helpful discussion that greatly helped us improve the manuscript. Before our point-by-point reply to his/her new comments, we would like to further emphasize the significance of our work:

1. Our work, for the first time, visualizes the gapped (100) surface state, making it possible to directly compare the experimental and calculated band structure to identify the HOTI phase of Bi_4Br_4 .
2. Our work, for the first time, present the spin texture of the (100) surface states, which provides vital information on confirming the non-trivial topology of Bi_4Br_4 .
3. Even if the observation of the hinge states is not so unambiguous as the surface states, we believe our results provide important insights into the electronic structure of Bi_4Br_4 . Compared with previous results, our observation of the gapped surface states with unprecedented data quality and its nice agreement with the theoretical calculation, is already a big advance towards the identification of the HOTI phase.

Nevertheless, we understand the reviewer's concern about the observation of the hinge states. In the following, we present new experimental results with further improved data quality to support our conclusion on the hinge states (see our reply below) and reply to the reviewer's comments point-by-point. We believe our new experimental evidence will largely relieve the reviewer's concern about the identification of the hinges states.

Reviewer's Comment:

The authors revised the manuscript following the referees comments. They added some additional information on how the experiment was performed and included in Supplementary Information a deconvolution of the ARPES data to support the identification of in-gap states on the (100) surface.

Authors' response:

We thank reviewer #2 for carefully reviewing our revised article and reply to the reviewers' comments below.

Reviewer's Comment:

The result is only marginally different from what was shown in Fig.3(d) of the main text. Without the guide-to-the-eye and the theoretical prediction of in-gap states it would be very difficult to identify these additional features from the ARPES data alone.

Authors' response:

With due respect, we cannot agree with the reviewer that the deconvoluted data is only marginally different from Fig. 3d of the main text. We understand the in-gap states in the deconvoluted data show much weaker intensity than the surface states, which is indeed similar as the raw data. We would like to remind the reviewer that the intensity of the hinge states is intricately extremely weak compared to the surface states. On top of this, the main difference between deconvoluted and raw data is that the non-trivial dispersion of the in-gap states is resolvable in the deconvoluted data even without the guide-to-eyes. Particularly, these dispersions cannot be simply explained by superposition of the surface band tails.

As the reviewer commented, there are two possible origins of the in-gap states in the ARPES data: the artificial intensity from the broadening of the surface states, or the hinge states. Even if the in-gap states are not so clear as the surface states, their non-trivial dispersion is sufficient to rule out the artifacts. Besides the non-trivial in-gap dispersions (please also see our new experimental results below), we provide several different methods to exclude the possibility of surface band tails (please see our previous response to the reviewer). Therefore, the only explanation for the observed in-gap states is the existence of hinge states.

Fig.R1 | Spectroscopic evidence for the existence of in-gap states. ARPES spectra after deconvolution to remove the spectral broadening effects. The arrows point to the hinge states. The measurements were conducted at around 30K.

In spite of this, we attach great importance to the reviewer's concern. To further improve the data quality, we conduct laser-based ARPES measurement at lower temperature (~ 30 K), as shown in Fig. R1. The dispersion of the in-gap states can now be clearly resolved without guide-to-eyes, in nice agreement with the expected hinge states (see our revised Supplementary Note 2). We believe this observation would

further relieve the reviewer’s concern about the artifacts of band tails and give more solid evidence on our identification of the HOTI phase. In the revised manuscript, we have updated Fig. 3 with new data.

Reviewer’s Comment:

The authors explain in the reply that the positions of the yellow bars marking the signature of in-gap states were defined by naked eyes, considering peaks and shoulders that are above the noise level. Clearly this is not a unique method to extract the dispersion: (i) there were other peaks and shoulders in the MDCs that the authors ignored; (ii) given that the MDCs were integrated in a certain energy window (reply to referee #3), one may wonder to what extent those shallow peaks were affected by the integration.

Authors’ response:

Fig. R2 MDC signature of the in-gap states. (a) stacking plot of MDCs in the gap region. The blue triangles, yellow bars, and gray circles mark the peak positions of the MDCs. The dashed lines are a guide of eyes for the band dispersion of the hinge states. (b) zoom-in plot of the MDCs in the gap region. (c) Zoom-in plot of the in-gap MDCs integrated in a smaller energy step. The open symbols represent the tails of the surface states. The dashed yellow bars represent peak that may be from the crossing point of two branches of the hinge states.

We appreciate the carefulness of the reviewer and agree with him/her that there were some peaks in the MDCs not marked. Some of the peaks without markers are from the tails of the surface states, and the noise of the data is not avoidable. In Fig. R2 (b), we add new markers for the peaks from the tails of the surface states. Although there are still peaks/features not marked, we would like to ask for the understanding that

the identification of the weak signal from the hinge states is indeed affected by the noise/background of data.

The integration of the MDCs curves helps exclude the peaks from noise/background. Due to the extremely weak intensity of the in-gap states, a smaller integration step will make it more difficult to identify the true signals from MDC peaks. Fig. R2(c) shows the MDCs integrated in a smaller energy step. The MDCs still show multi-peak structures and the dispersion of the in-gap states can be tracked, though it is much more difficult.

Although we believe our MDC analysis properly evidences the existence of the hinge states, we fully understand the concern of the reviewer and agree that this method is not strictly accurate (due to the extremely weak intensity of the hinge states and the effect of data noise and background). Therefore, we removed the MDC curves in the main text and present our new data with clear dispersion of the hinge states in the revised manuscript.

Reviewer's Comment:

I don't think that the authors could have done much better with the existing data set, but I have the impression that the "compelling evidence" for hinged states isn't actually that strong.

Authors' response:

With our point-by-point reply and new experimental results, we believe that we observe the dispersion of the hinge states and evidently rule out the artifacts of band broadening. Together with the observation of the (100) gapped surface states and non-trivial spin texture, we believe our work present "compelling evidence" for the HOTI phase. We sincerely ask the reviewer to reconsider his/her recommendation and look forward to his/her further comments.

Response to Reviewer #3:

Reviewer's Comment:

The authors have carefully considered the comments and questions from the reviewer. Most of the unclear statements or conclusions in the first version of the manuscript have been clarified by additional figures and discussions. The results are certainly at the experimental limit at the moment and ask for further experimental investigations in the future. However, the paper represents a significant step towards a better understanding of this interesting material. I recommend to publish it in Nat. Commun. in its present form.

Authors' response:

We thank reviewer #3 for acknowledging that “the unclear statements or conclusions in the first version of the manuscript have been clarified” and “the paper represents a significant step towards a better understanding of this interesting material”. We appreciate his/her recommendation for the publication of our work.

List of changes

1. We added “These in-gap states can be better resolved in the ARPES spectra after deconvolution, which is a commonly used method to remove the spectral broadening effect (Fig. 3g, h). Notably, a double-crossing-shaped dispersion is shown clearly inside the gap and agrees well with the calculated hinge states in Fig. 3a.” in line 17, page 5.
2. We replaced Fig. 3g by new experimental results to show the non-trivial dispersions of the in-gap states and updated the captions.

REVIEWERS' COMMENTS

Reviewer #2 (Remarks to the Author):

The authors took my concerns in consideration: they removed the MDCs analysis and included in the manuscript a deconvolution of ARPES data measured at lower temperature, where the signature of hinged state is stronger. Based on this new evidence, I recommend publication of the manuscript.

However, I would like to point out an inconsistency between the changes described in the rebuttal and the current version of the manuscript.

In the rebuttal the authors write that they conduct laser-ARPES measurement at lower temperature and updated Fig.3 with the new data.

However in Fig.3 they only added a panel with the deconvoluted data measured at 30 K but did not change the data in panels (c-f)) that were presumably measured at higher temperature.

The ARPES data at 30 K before convolution are not shown anywhere.

I think this should be corrected before publication, but I don't need to review the manuscript again.

Response to Reviewer #2:

Reviewer's Comment:

The authors took my concerns in consideration: they removed the MDCs analysis and included in the manuscript a deconvolution of ARPES data measured at lower temperature, where the signature of hinged state is stronger. Based on this new evidence, I recommend publication of the manuscript.

Authors' response:

We thank reviewer #2 for helping us improve the manuscript. We are grateful to him/her for recommending the publication of our manuscript.

Reviewer's Comment:

However, I would like to point out an inconsistency between the changes described in the rebuttal and the current version of the manuscript.

In the rebuttal the authors write that they conduct laser-ARPES measurement at lower temperature and updated Fig.3 with the new data.

However in Fig.3 they only added a panel with the deconvoluted data measured at 30 K but did not change the data in panels (c-f) that were presumably measured at higher temperature.

The ARPES data at 30 K before convolution are not shown anywhere.

I think this should be corrected before publication, but I don't need to review the manuscript again.

Authors' response:

We thank the reviewer for his/her suggestion. We kept ARPES data at 80 K in Fig. 3 for better comparison between the experiment and calculation since the data shows more information of the conduction band above the Fermi level after being divided by the Fermi-Dirac function.

We thank the reviewer for suggesting us showing the data without deconvolution. In the revised manuscript, we have added the ARPES data at 30 K in the Supplementary Information following the reviewer's suggestion.